

# TTECCDU: a blockchain-based approach for expressive authorization management

Uzma Mahar[1], Muhammad Aleem[1] and Ehtesham Zahoor[2]

[1] National University of Computer and Emerging Sciences, Islamabad, Pakistan
[2] Educative, Inc, Islamabad, Pakistan

## ABSTRACT

Authorization uses the access control policies to allow or limit a user the access to a resource. Blockchain-based access control models are used to manage authorization in a decentralized way. Many approaches exist that have provided the distributed access control frameworks which are user driven, transparent and provide fairness with its distributed architecture. Some approaches have used authorization tokens as access control mechanisms and mostly have used smart contracts for the authorization process. The problem is that most of the approaches rely on a single authorization factor like either trust or temporal; however, none has considered other important factors like cost, cardinality, or usage constraints of a resource making the existing approaches less expressive and coarse-grained. Also, the approaches using smart contracts are either complex in design or have high gas cost. To the best of our knowledge, there is no approach that uses all the important authorization factors in a unified framework. In this article, we present an authorization framework: TTECCDU that consists of multi-access control models *i.e.*, trust-based, cost-based, temporal-based, cardinality-based, and usage-based to provide strong and expressive authorization mechanism. TTECCDU also handles the delegation context for authorization decisions. The proposed framework is implemented using smart contracts which are written in a modular form so that they are easily manageable and can be re-deployed when needed. Performance evaluation results show that our smart contracts are written in an optimized manner which consume 60.4% less gas cost when the trust-based access is compared and 59.2% less gas cost when other proposed smart contracts from our approach are compared to the existing approaches.

# INTRODUCTION

Security issues in large scale distributed systems have gained wide attention of researchers. An expressive authorization management system is needed to provide the security support which protects digital resources on the internet. The access of any particular resource should only be granted to the entities that hold the actual rights. Access control systems are meant for this purpose, as they enable only authenticated subjects to access the resources through access control policies. These access control policies check the access context every

Corresponding authors
Uzma Mahar,
uzma.mahar@nu.edu.pk
Muhammad Aleem,
m.aleem@nu.edu.pk

time a request for access is made and returns to the subject the access decision as access granted or denied.

Wide range of access control models have been presented in the scientific literature. The classic access control models include the discretionary access control (DAC) model (*Downs et al., 1985*; *Jayant et al., 2014*) which does not cope well with large scale distributed systems as it is impossible to make an access control list for everyone in the system. Another classical model is mandatory access control (MAC) model (*Bell & La Padula, 1976*; *Denning, 1976*) which is system enforced access control but this approach lacks flexibility and is difficult in implementing and programming. Limitations to these models led to role based access control (RBAC) model (*Rahman et al., 2020*). The RBAC is an access control model that associates permissions with roles, and then assigns users to suitable roles based on their capabilities and responsibilities provided by the organization. In user based access control (UBAC) model (*Rahman et al., 2020*), the policies are written for individual users. To manage increasing number of users and writing policies for each of them is a difficult and time consuming task. Hence, we have used RBAC model in our proposed approach. RBAC uses the concept of roles where roles are assigned different permissions to access a resource. Users are added to each role depending on its responsibilities assigned by any organization. The advantages of using RBAC is that a complete hierarchical structure of permissions is introduced and provides least privilege that means users are associated to different roles to perform tasks.

Access control systems based on centralization (*Downs et al., 1985*; *Jayant et al., 2014*; *Bell & La Padula, 1976*; *Denning, 1976*) are not flexible and do not provide an expressive authorization mechanism. To address the issue of authorization, access control systems based on decentralization are needed. For decentralized access control, our article focuses on blockchain based authorization management. Blockchain is point of discussion that has gained wide attention from academia and industries in recent years. Blockchain works on the decentralization concept, is distributed and tamper-proof (*Elrom, 2019*; *Tschorsch & Scheuermann, 2016*) ledger which is provided to all the nodes in the chain. Due to decentralization in blockchain, every node in the chain can validate the transactions whereas in centralized systems, only the administrator could do it. Blockchain technology is widely used because of its features (*Xu et al., 2017*), like decentralization as stated above, change resistance and traceability. The existing blockchain based authorization management systems are not expressive. To the best of our knowledge, there is no approach that uses all the important authorization factors like trust, temporal, cost, cardinality, and usage in one framework which makes the existing proposals coarse-grained.

There are approaches that claim to provide fine grained authorization mechanism but they have overlooked some of the most important factors like trust, cost, cardinality, usage and temporal constraints. Trust factor is important to evaluate the trust score of user and maintain/update that trust score for the grant or denial of access to the resource requested. Similarly, temporal constraints enhance the expressiveness of the authorization management process as the time slots can be checked to provide the access of a resource. Other aspects like how many users can get an access of a resource or how many times a particular user can get an access of a resource at maximum are also equally important

factors for an accurate authorization decision. Hence, a framework is needed that combines all the above important factors for a fine grained authorization mechanism. Our approach TTECCDU *i.e.,* trust-based, temporal-based, cost-based, cardinality-based, and usage-based focuses on decentralized access control models . Combination of these access control models in one framework provides an expressive authorization management system. Our proposed approach also handles the delegation context for authorization decisions.

We have also presented a case study as a motivating example. A journalist who wants to upload a news article on news blockchain needs to be checked if he/she meets the trust criteria set by the system. A reader who wants to read the news article also needs to be verified using different factors. Our framework's trust-based access control model can be used for the verification of journalist and the other proposed models *i.e.,* cost-based, temporal-based, cardinality-based, and usage-based can be used for the verification of the reader. Our framework TTECCDU can also be used to check the delegation constraints, if any. Our framework having the combination of multi access control models is implemented using the smart contracts. Smart contracts design is modular so that they are easily manageable and can be re-deployed when needed. The data structure used in smart contracts are used in a way that minimizes the gas cost which is evident from the performance evaluation results. Our core contributions are summarized as follows:

1. TTECCDU combines trust, cost, temporal, cardinality and usage access control models to present a fine grained and more expressive authorization management process.
2. TTECCDU access control models are implemented using smart contracts which are written in a modular form so that they are easily manageable and can be re-deployed when needed.
3. Delegation is important for an accurate authorization decision. Our proposed approach also handles the delegation context for an expressive authorization management.
4. We have performed comprehensive performance and security analysis. Results show 60.4% gas cost reduction in case of trust-based access and 59.2% while comparing rest of the approach.

The rest of the article is structured as follows: we have presented the related work in Section 2 and a motivating example of news blockchain in Section 3. We have then presented the proposed approach in Section 4 and then detailed the authorization management process in Section 5. Section 6 presents the security analysis of our approach. Section 7 shows performance evaluation results and Section 8 concludes the article.

## RELATED WORK

One of the major domains for research these days is authorization policies management and one of the sub domains is access control models designed to implement the authorization management process. Recent research shows that blockchain based access control systems can mitigate the problems of centralized authorization schemes and can handle the security and performance issues to a greater extent. *Maesa, Mori & Ricci (2018)*, presented a blockchain based access control system where they have published the policies publicly to prevent fraudulent denying of the rights granted by an enforceable policy. However,

the approach has performance overheads like delayed response time and increased gas cost. *Zyskind, Nathan et al. (2015)*, presented a personal data management platform focused on privacy where the authors claim to provide data ownership, data transparency, and auditability with fine grained access control. However, a detailed security analysis of the approach is missing and the system may result in performance issues w.r.t to high latency and excessive energy consumption. *Ouaddah, Abou Elkalam & Ait Ouahman (2016)* proposed a distributed access control framework based on blockchain technology named FairAccess. The approach is user driven, transparent and provides fairness with its distributed architecture but the access control framework is coarse-grained and is less expressive. The tokens are not implemented using smart contracts and locking scripts have a limited computing capability. *Es-Samaali, Outchakoucht & Leroy (2017)* proposed a blockchain based access control framework which addresses security and privacy issues in big data. They have used authorization tokens as an access control mechanism, delivered through emergent cryptocurrency solutions. The drawback of the approach is that the framework uses no specific access control model for authorization and the part of the approach still relies on centralized AMP for authorization of a resource.

Much access control work is specifically done for IoT systems. *Khalid et al. (2020)*, presented a delay-sensitive blockchain-enabled access control mechanism for IoT systems. The strengths of the approach are that it handles security problems well and allows communication between devices from different groups. However, the approach incurs communication overhead and performance overhead in terms of latency. *Zhu et al. (2018a)*; *Zhu et al. (2018b)*; *Ourad, Belgacem & Salah (2018)*, proposed a lightweight solution using blockchain to run on any system developed using the Ethereum framework. They devised an authorization solution which verifies user identity through a smart contract and grants access to the blockchain, further persisted by smart contract tokens. The strengths of these approaches are that they are user driven, transparent, with fairness and distributed architecture. However, they have an integrated design structure, performance overhead in terms of latency, more gas cost consumption. Also, access control is limited to attributes and transactions; other dimensions like trust, cost, temporal and usage constraints are not considered. *Novo (2018)* proposed a scalable access control management platform using smart contracts. The approach performs better even when the miners are increased, but the drawback is that all the access control operations are performed in a single smart contract which is not an optimized solution and hence reduces flexibility of the approach. *Outchakoucht, Hamza & Leroy (2017)* proposed an approach which applies reinforcement learning to dynamically update the policies. The feedback of the transactions is used to update policies dynamically which improves the integrity, trust level, and credibility but the approach may increase computational complexity due to large number of devices. *Hwang, Choi & Kim (2018)* proposed a method to facilitate data transfer requests between authenticated devices. The approach increased scalability and usability but is not expressive as the access control scheme does not consider some important authorization factors like trust, time, cost and usage. *Zhang et al. (2018)* developed a smart contract-based access control management system including a judge contract and a register contract to achieve decentralized and secure access control for IoT devices. The authors have used

multiple smart contracts for the approach including the trust evaluation mechanism which makes the approach expressive. However, the gas cost consumption is high due to the complexity of smart contracts structure. They have also taken time as a performance evaluation measure which is not a good criteria for results evaluation.

Authors have also worked on blockchain based authorization mechanisms based on attribute-based access control model. *Ye et al. (2014)* presented an authentication and access control scheme, based on attribute-based access control model. The approach provides solution to resource constrained problem in the perception layer of internet of things. However, using attribute-based access control model requires defining and implementing all of the attributes which can be time consuming when the attributes increase. *Alansari et al. (2017)* presented a blockchain based authorization mechanism to allow federated organizations to enforce attribute-based access control policies on their data in a privacy-preserving fashion. Their approach allows organizations to control which users from other federated organizations can access which data however, there are some drawbacks of the approach. Managing thousands of attributes is a time-consuming task, the approach requires more effort and resources to design policies and incurs performance overhead in terms of latency. *Maesa, Mori & Ricci (2019)* proposed an approach that maintains both the subject attributes and access control policies using smart contracts. Both the policies and the required attributes for the evaluation of policies are auditable on the blockchain. However, the values of the attribute are not editable and can only be modified with a new transaction on the attributes blockchain increasing the encryption computations. *Wang, Wang & Zhang (2019)* presented a decentralized cloud storage framework using an attribute-based cipher-text policy. Ethereum smart contracts are used to enforce the trust. Furthermore, the authors considered an expiration time for the smart contracts which restricts the user to access data only in a specific time frame. The drawback of the approach is that it cannot reach high throughput and cannot perform well in an environment with many concurrent requests. *Dramé-Maigné, Laurent & Castillo (2019)* proposed an approach to dynamically manage multi-endorsed attributes and trust anchors. Its focus on attributes offers flexibility, expressiveness, and user-centricity, accommodating the dynamic addition of subjects. A list of trusted entities and miners is provided but no in-depth trust evaluation mechanism is given to evaluate the trust of entities at run time. Some authors like *Rahman et al. (2020)* also used role-based access control model for authorization management. They proposed a context-aware, auditable and dynamic role-based access control using Ethereum smart contracts. The approach is user-centric, context aware and dynamic but there are some limitations of the approach as well. The approach is coarse-grained and less expressive as other factors like trust, time, cost, and usage are not considered for authorization. Also, the user device cannot trace the location of the user in case of no internet connectivity.

Some authors have proposed blockchain based authorization management for healthcare platforms as well. *Tanwar, Parekh & Evans (2020)* proposed an access control policy algorithm for improving data accessibility between healthcare providers. Performance metrics in blockchain networks, such as latency, throughput, Round Trip Time (RTT), have been optimized through the approach but the transaction cost can be increased as

the approach uses smart contract to execute all the inward transactions for health records. Some authors have tried to make the access control schemes cost efficient. *Vora et al. (2018)* presented a blockchain-based framework for efficient storage and maintenance of EHRs. A database manager entity for generating a link to the patients' EHRs in local databases is used by the authors which has addressed the data storage issue of blockchain. However, their approach uses five smart contracts which consume more gas cost due to their structure. *Wiraatmaja et al. (2021)* proposed a layered BBAC architecture by combining blockchain with blockchain oracle and tamper-proof decentralized storage (*e.g.*, IOTA). The three-layered architecture achieves robust, auditable, and cost-efficient access control by migrating the meta data from the blockchain to the decentralized storage. However, a lot of work is done off chain. The attributes management is still done by the attribute manager which makes this part of the approach centralized and additional fetching from the decentralized storage to retrieve the address table is required which increases the time to finish one access request. *Steichen et al. (2018)* proposed a modified version of the Inter Planetary File system (IPFS) that leverages Ethereum smart contracts to provide access-controlled file sharing. The approach can efficiently store and share large files because of using resource hashes but the usage of smart contract to maintain the access control lists causes computational overhead and performance overhead in terms of latency.

Trust is an important factor to be considered for authorization management in blockchain systems. *Di Pietro et al. (2018)* proposed a trust-based access control system where trust is evaluated using a credit system. The trust is managed and updated dynamically and the approach enables registration of new users and maintaining their record based on user behavior. However, the approach is not expressive as it does not consider other important factors like cost, temporal, cardinality and usage constraints. Also, the system using credit system is vulnerable to malicious activities. *Oualhaj et al. (2020)* presented a blockchain based decentralized trust management model and an algorithm to detect malicious nodes and to select miners. The strength of the approach is that the authors have managed to reduced energy cost and transaction costs in case of network saturation. However, the trust updating algorithm only considers the honesty value as a trust factor and ignores other important trust factors like usage history of node. Also, the trust value calculation for a new node that enters the system is not considered. *Yang et al. (2018)* proposed a decentralized trust management system in vehicular networks based on blockchain technique. The credibility of messages received from the neighboring nodes can be verified using the Bayesian Inference Model. Their approach reduces the communication and computation overheads but there are some limitations as well. The trust management of all the vehicles is done *via* RSUs. In case of a compromised RSU, the system will not function accurately. Also, the trust mechanism only considers ratings as a trust factor and ignores other important factors which makes the approach less expressive. *Putra et al. (2021a)* worked on a decentralized attribute-based access control mechanism with an auxiliary Trust and Reputation System (TRS) for IoT authorization. The approach is resilient against attacks like bad mouthing, sybil and newcomer attacks, and also handles illegitimate attempts to increase reputation scores. However, the approach has bootstrapping problem which means that a new node with zero reputation score cannot

participate in the network. *Singh et al. (2020)* proposed a blockchain-based decentralized trust management scheme using smart contracts. The authors use blockchain sharding which reduces the load on the main blockchain and increases the transaction throughput. However, sharding can be risky in case several nodes suddenly go offline or if the network grows so large that a full copy of transaction history is no longer maintained by anyone in the network.

Authors have also considered delegation aspect to make the authorization policies management more expressive. *Xu et al. (2018)* proposed a federated capability-based delegation model (FCDM) to support hierarchical and multi-hop delegation. The benefits of the approach are that JSON tokens are used which have less overhead as compared to CAP tokens and the approach is scalable when the network size increases. However, each node has to synchronize its local chain with the main blockchain which causes delays in case of poor network connectivity and increases the overhead for each device. *Sherazi et al. (2022)* presented an approach that decouples authorization logic from the core capabilities of a smart contract and provides contextual delegation in case of emergency situations. The approach handles advance conflict handling, redundancy reduction, and advance contextual delegation. However, it ignores other important factors like trust, temporal, cost, usage and cardinality. It also consumes more gas cost. The abbreviations of authorization types are given in Table 1 and an overview of the related works is given in Table 2.

The research gap is that the existing approaches that provide authorization management are not expressive, (*Steichen et al., 2018*; *Zhu et al., 2018a*; *Zhu et al., 2018b*; *Ourad, Belgacem & Salah, 2018*; *Di Pietro et al., 2018*; *Hwang, Choi & Kim, 2018*). Most of the approaches rely on a single authorization factor like either trust or temporal but none has considered other important factors like cost, cardinality, or usage constraints of a resource, *Rahman et al. (2020)* and *Sherazi et al., 2022*). To the best of our knowledge, there is no approach that uses all the important authorization factors like trust, temporal, cost, cardinality, and usage in one framework which makes the existing proposals coarse grained and less expressive. The existing approaches that use trust as an authorization factor provide list of trusted entities and miners but no in-depth trust evaluation mechanism is given to evaluate the trust of entities at run time, (*Dramé-Maigné, Laurent & Castillo, 2019*; *Di Pietro et al., 2018*; *Oualhaj et al., 2020*; *Yang et al., 2018*; *Putra et al., 2021a*; *Singh et al., 2020*). Mostly have bootstrapping problem *i.e.,* new node with zero reputation score cannot participate in the network, (*Oualhaj et al., 2020*; *Putra et al., 2021a*). Most of the authors that claim to provide complete decentralized authorization schemes also have a part of the approach based on centralization either for attributes management or for the authorization of a resource, (*Es-Samaali, Outchakoucht & Leroy, 2017*; *Yang et al., 2018*; *Wiraatmaja et al., 2021*). The existing approaches use smart contracts but either they have a very complex design which makes the proposal inflexible or they are written in a way which consumes high gas cost, (*Zhu et al., 2018a*; *Zhu et al., 2018b*; *Ourad, Belgacem & Salah, 2018*; *Novo, 2018*; *Zhang et al., 2018*; *Tanwar, Parekh & Evans, 2020*; *Vora et al., 2018*; *Sherazi et al., 2022*).

**Table 1  List of authorization types used in the literature.**

| Authorization type | Full form |
| --- | --- |
| XACML | Extensible Access Control Markup Language |
| ABAC | Attribute Based Access Control |
| ACL | Access Control List |
| CapBAC | Capability Based Access Control |
| RBAC | Role Based Access Control |
| BBAC | Blockchain Based Access Control |
| TBAC | Trust Based Access Control |

## CASE STUDY—NEWS BLOCKCHAIN

As a motivating example we have considered the case of a news blockchain where different roles can perform different tasks after going through the authorization mechanism. Large files and resources cannot be stored on blockchain due to storage space issues. IPFS is one of the implementations of DFS which provides us cryptographic hashes of any resource and we can just store these hashes on blockchain instead of complete files (_Steichen et al., 2018_). The news articles are stored on IPFS nodes using smart contracts written in Ethereum. For the news blockchain as shown in Fig. 1, one role can be of a journalist which is assigned to a user named Bob who wants to get an access to publish a news article (can be any resource or a file in general) let us say article X on the news blockchain and the other role can be of a reader assigned to a user Alice who wants an access to read article X. When journalist Bob make an access request to publish article X on news blockchain, the authorization model checks the trust score of the Bob. The news blockchain already maintains a trust limit and the trust score of Bob is compared to the trust limit provided by the system. If Bob has a trust score more than the specified trust limit, he will get the access to the news blockchain and can upload the article X. This whole process can be done using the trust smart contract; the details of which are provided in the proposed methodology section.

Now, let us take into consideration the reader's perspective as shown in Fig. 2. Let us say the role _i.e.,_ reader is assigned to a user Alice who wants to gets an access to the article X. Our system provides the authorization decision as per the context with the help of smart contracts discussed in Section 4. There are four basic contexts to be handled as per the four contracts _i.e.,_ cost contract, temporal contract, cardinality contract and usage contract.

### Context 1: Cost based access control
Alice pays a certain amount for the article and gets an access.

### Context 2: Temporal based access control
For a certain time span like 3 pm to 5 pm, the access can be provided to the reader John from role A, and for the other time span say 5 pm to 7 pm, the access can be provided to reader Alice from role B.

**Table 2  An overview of related works.**

| Proposal | Authorization type | Limitations |
|---|---|---|
| *Maesa, Mori & Ricci (2018)* | XACML | Delayed response time, increased gas cost |
| *Zyskind, Nathan et al. (2015)* | ABAC | High latency, excessive energy consumption |
| *Ouaddah, Abou Elkalam & Ait Ouahman (2016)* | ABAC | Coarse-grained access mechanism |
| *Es-Samaali, Outchakoucht & Leroy (2017)* | ACL | Centralized AMP for authorization of resource |
| *Khalid et al. (2020)* | ACL | Communication overhead, high latency |
| *Zhu et al. (2018a)* | CapBAC | Coarse-grained, high gas cost |
| *Zhu et al. (2018b)* | CapBAC | Coarse-grained, high gas cost |
| *Ourad, Belgacem & Salah (2018)* | CapBAC | Coarse-grained, high gas cost |
| *Novo (2018)* | ACL | Less flexible |
| *Outchakoucht, Hamza & Leroy (2017)* | ABAC | Increased computational complexity |
| *Hwang, Choi & Kim (2018)* | ABAC | Less expressive; important authorization factors missing |
| *Zhang et al. (2018)* | ABAC | High gas cost |
| *Ye et al. (2014)* | ABAC | Management of thousands of attributes is time consuming |
| *Alansari et al. (2017)* | ABAC | Requires more effort and resources to design policies |
| *Maesa, Mori & Ricci (2019)* | ABAC | Values of the attributes are not editable |
| *Wang, Wang & Zhang (2019)* | ABAC | Cannot reach high throughput |
| *Dramé-Maigné, Laurent & Castillo (2019)* | ABAC | No in-depth trust evaluation mechanism at run time |
| *Rahman et al. (2020)* | RBAC | Less expressive; important authorization factors missing |
| *Tanwar, Parekh & Evans (2020)* | ACL | Increased transaction cost |
| *Vora et al. (2018)* | ABAC | High gas cost |
| *Wiraatmaja et al. (2021)* | BBAC | Centralized AM for attributes management |
| *Steichen et al. (2018)* | ACL | Computational overhead, high latency |
| *Di Pietro et al. (2018)* | TBAC | Less expressive; important authorization factors missing |
| *Oualhaj et al. (2020)* | TBAC | Bootstrapping problem |
| *Yang et al. (2018)* | TBAC | Compromised RSU can result in faulty decisions |
| *Putra et al. (2021a)* | TBAC | Bootstrapping problem |
| *Singh et al. (2020)* | TBAC | Sharding has risks when network grows |
| *Singh et al. (2020)* | CapBAC | Delays in case of poor network connectivity |
| *Singh et al. (2020)* | RBAC | Less expressive; important authorization factors missing |

## Context 3: Cardinality based access control

Suppose if the resource access limit = 100 and the number of users granted the access of resource is less than the cardinality limit *i.e.,* 100, the access will be granted else denied.

## Context 4: Usage based access control

Let us say the predefined access limit for article X is 10, if Alice makes an access request more than 10 times, the access will be denied.

In Fig. 3, we have shown the basic implementation of IPFSResourceContract where the concept of hashing is shown as evident from line number 8 to 24. There are four functions shown *i.e.,* addResource (line 8), updateResourceHash (line 12), deleteResource (line 17), and getResource (line 22). Any user can add resources to IPFS and store the respective hashes on blockchain through the IPFSResourceContract but only resource owner can update, get and delete the resource. It can be seen that IPFSResourceContract has some

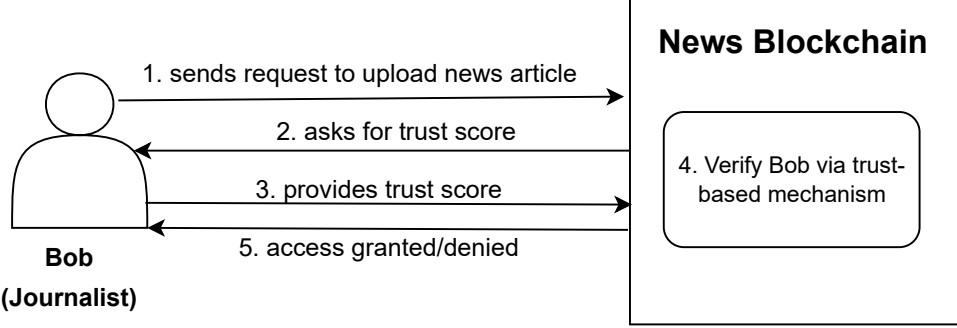

**Figure 1** Article uploading scenario on news blockchain.

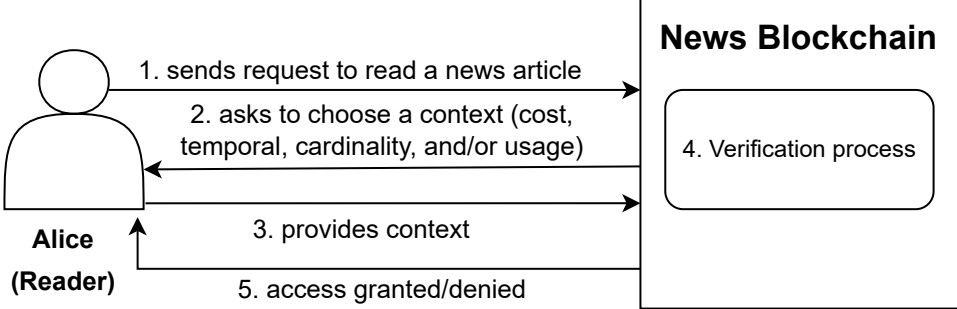

**Figure 2** Accessing an uploaded article scenario on news blockchain.

basic level of authorization but it still needs some advance authorization for better security. The contract here is only user-based and managing policies for every user is a difficult task. Another problem, as seen in line number 10 of Fig. 3, is that any user can add resource. We need to convert the access from user-based to roles-based model so that roles can be assigned to users and only roles need to be managed instead of managing policies for every single user.

# PROPOSED METHODOLOGY

This section provides a detailed overview of our proposed framework TTECCDU that consists of five access control models and delegation which is used for a secure and efficient authorization management. The system modelling, proposed architecture, the algorithms, the smart contracts working and interaction, and the delegation details are presented in this section.

## Formalization of authorization and delegation process

The authorization process provides the access decision based on the access defined policies keeping in consideration the delegation aspect. The approach can be formalized using a range of sets. The set R is a set of roles to whom access has been granted or denied,

```
1 ∨ contract IPFSResourceContract {
2 >     struct Resource{ ⋯
6     }
7     mapping(string=>Resource) resources;
8 ∨   function addResource(string calldata _hash,string calldata _name) public {
9         resources[_name].hash = _hash;
10        resources[_name].rowner = msg.sender;
11    }
12 ∨  function updateResourceHash(string calldata _name, string calldata _hash) public {
13        require(msg.sender==resources[_name].rowner, "user is not allowed");
14        resources[_name].hash = _hash;
15    }
16
17 ∨  function deleteResource(string calldata _name) public {
18        require(msg.sender==resources[_name].rowner, "user is not allowed");
19        delete resources[_name];
20    }
21
22 ∨  function getResource(string calldata _name) public view returns( string memory _hash, address _rowner){
23        _hash = resources[_name].hash;
24        _rowner = resources[_name].rowner;
25    }
```

**Figure 3  IPFSResourceContract with basic authorization.**

$R = \{r_1, r_2, \ldots., r_n\}$. Set U is set of users to whom roles are assigned, $U = \{u_1, u_2, \ldots., u_n\}$. Each role contains a set of users $i.e.$, $u_1 \in r_1, u_2 \in r_2, \ldots.u_n \in r_n$. So, we can say that $U \subset R$. The set Res is set of resources whose access is granted or denied, $Res = \{res_1, res_2, \ldots., res_n\}$. The set T is set of tasks that a user can perform on a resource, $T = \{t_1, t_2, \ldots., t_n\}$. Set D is set of authorization decision regarding the access of a resource, $D = \{grant, deny\}$. All possible policies of the system are in set P as shown in Eq. (1). P contains defined access policies p which can be represented as $p = \{r_i, res_i, t_i, d_i\}$ where $r_i \in R$, $res_i \in Res$, $t_i \in T$, and $d_i \in D$.

$$P = R \times Res \times T \times D. \tag{1}$$

Our proposed approach contains the defined policies for access control which can be represented as DP where $DP \in P$. For a user $u_i$ who is assigned some role $r_i$ to have access of $res_i$, the policy p should be part of DP where access decision $d_i$ is 'grant' and $t_i$ defines the action that can be taken. If $u_1 \in r_1$ delegated his access privileges to $u_n \in r_n$, then the policy $p(r_n, res_i, t_i, d_i)$ can be added to DP.

## Authorization policies management using multi-access control models

Our proposed approach TTECCDU is based on an access control mechanism that considers the usage of smart contracts on ethereum. The access granted or denied from the request owner to the resource requestor will be based on the following factors: trust-based access control, cost-based access control, temporal-based access control, cardinality-based access control, and usage-based access control, after checking the delegation aspect. An overview of the proposed blockchain based architecture is shown in Fig. 4. The process starts when the policy designer(s) deploys the smart contracts on Ethereum blockchain as shown in step 1 of Fig. 4. Access Control Contract is the core contract that handles all the contexts and provides authorization decisions to users by invoking other smart contracts including

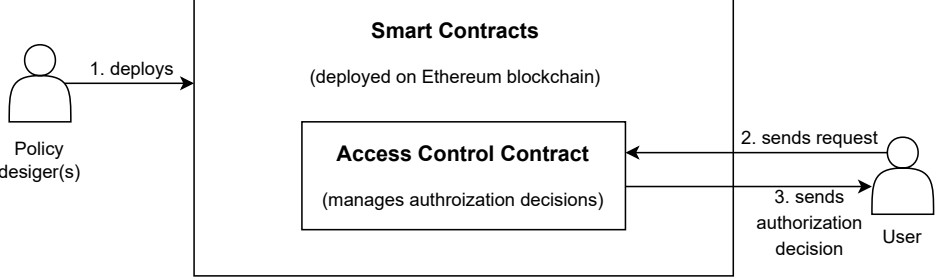

**Figure 4** Blockchain based proposed architecture.

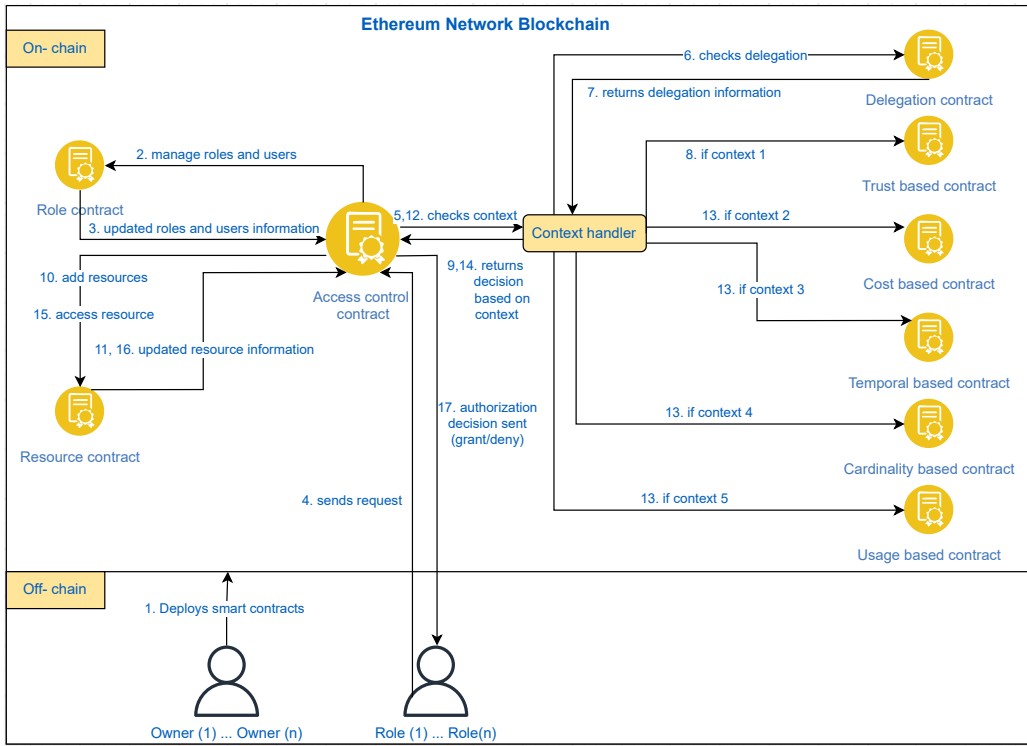

**Figure 5** Working flow of the proposed framework TTECCDU.

the delegation contract as shown in steps 2,3 and 4 of Fig. 4. The detailed working flow of the architecture is shown in Fig. 5.

In Fig. 5, at first, the policy designer(s) deploy the smart contracts on ethereum blockchain as shown in step 1 of Fig. 5. The proposed approach is divided into multiple smart contracts for the modular approach and easy management of the process. The major smart contract is the 'access control contract' that handles all the requests from users and makes authorization decisions using other smart contracts. The roles and users management is done through the role contract and the updated information is sent back to the access control contract as shown in steps 2 and 3, respectively. A user who is assigned some role let's say role 'A'

makes a request to add a resource. This request is received by the access control contract as shown in step 4 of Fig. 5. The access control contract makes the authorization decision by checking the contexts and invoking the smart contracts accordingly as shown in step 5. The access control contract first invokes the delegation contract and checks the delegation aspect. After receiving the delegation information *i.e.,* steps 6–7, the contract check the context through context handler. If context is equal to 1, the trust based contract is invoked as shown in step 8. The access decision is sent back to the access control contract in step 9. If the decision is 'grant', then the resource contract is invoked. The resource contract manages the resources that are stored on IPFS and the hashes of those resources are stored on blockchain through this contract. The resource is added and the updated information is sent back to the access control contract as shown in steps 10 and 11.

The other perspective of the proposed approach is of accessing an existing resource through the access control contract. The user send an access request which is received by the access control contract as shown in step 4 of Fig. 5. The access control contract checks the contexts through context handler, step 12. The access control contract first invokes the delegation contract and checks the delegation aspect where steps 6 and 7 are repeated. The context handling is done through step 13 where if context is equal to 2, cost based contract is invoked, if context is equal to 3, temporal based contract is invoked, if context is equal to 4, cardinality based contract is invoked, and if context is equal to 5, usage based contract is invoked. The contextual decision is sent back to the access control contract *i.e.,* step 14. Access control contract invokes the resource contract to access the resource if the decision is 'grant' and resource information is received, steps 15-16. In step 17, the authorization decision is sent back to the user.

The authorization process of our methodology can be elaborated through different algorithms. Algorithm 1 shows the access control mechanism to upload a resource using trust-based and cost-based access control. Line 1 of Algorithm 1 shows the procedure names. Line 2 checks if the user belongs to the role, line 3 checks the delegation constraints, line 4 and 13 checks the contexts. If context is 1 *i.e.,* trust based access is required, then lines 5 to 12 of Algorithm 1 checks if the trust score of the user who wants to upload the resource is greater or equal to the trust limit set by the system, then the access is granted, else denied. Similarly, if context is 2 *i.e.,* cost based access is required, then lines 14 to 21 of algorithm 1 implements the policy *i.e.,* if the amount provided by the user is equal to the cost of the resource, the access is granted, else denied. Line 24 ends the procedure.

---

**Algorithm 1** Upload resource using trust based and cost based access control models

---

**Require:** Contexts must be defined

1: **procedure** *uploadresource* () , *accesspaidresource* ()
2:    **if** $u_n \in r_n$ **then**
3:       **if** $\neg$ isdelegated() **then**
4:          **if** *context* $= 1$ **then**
5:             **if** trust_score of $u_n \geq$ trust_limit **then**
6:                $d_i =$ grant
7:                $D = d_i$
8:             **else**
9:                $d_i =$ deny
10:                $D = d_i$
11:             **end if**
12:          **end if**
13:          **if** *context* $= 2$ **then**
14:             **if** amount given by user = cost of resource **then**
15:                $d_i =$ grant
16:                $D = d_i$
17:             **else**
18:                $d_i =$ deny
19:                $D = d_i$
20:             **end if**
21:          **end if**
22:       **end if**
23:    **end if**
24: **end procedure**

---

Algorithm 2 shows the access mechanism of accessing a resource using temporal based, cardinality based, and usage based access control models. Line 1 of algorithm 2 shows the parameters taken by the function accessresource() are: name of the role who wants to access the resource and name of the resource for which access is required. On successful authorization process, the hash of the resource is returned to the user. Line 2 checks if the user belongs to the role, line 3 checks the delegation constraints, line 4, 13, and 22 checks the contexts. If context is 3 *i.e.,* temporal based access is required, then lines 5 to 12 of algorithm 2 checks that if the user makes an access request at a time that is above the pre-defined starting time and below the pre-defined end time, the access is granted, else denied. Line 13 of algorithm 2 checks if context is 4 *i.e.,* cardinality based access is required, then lines 14 to 21 checks if the number of users granted the access of resource is less than or equal to the resource access limit, the access will be granted else denied. Line 22 of algorithm 2 checks if context is 5 *i.e.,* usage based access is required, then lines 23 to 30 check if number of permissions assigned to a user to access a resource is less than or

equal to the predefined resource access limit, then the access will be granted, else denied. Line 33 ends the procedure.

---

**Algorithm 2** Access resource using temporal based, cardinality based, and usage based access control model

---

**Require:** Contexts must be defined

  1: **procedure** *accessresource*(role, resource_name) **returns** (resource_hash)

  2:    **if** $u_n \in r_n$ **then**

  3:       **if** ¬ isdelegated() **then**

  4:         **if** *context* = 3 **then**

  5:           **if** block.timestamp $\geq$ start_time of resource && block.timestamp $\leq$ end_time of resource **then**

  6:             $d_i$= grant

  7:             $D$= $d_i$

  8:           **else**

  9:             $d_i$= deny

10:             $D$= $d_i$

11:           **end if**

12:         **end if**

13:         **if** *context* = 4 **then**

14:           **if** number of resources accessed $\leq$ resource access limit **then**

15:             $d_i$= grant

16:             $D$= $d_i$

17:           **else**

18:             $d_i$= deny

19:             $D$= $d_i$

20:           **end if**

21:         **end if**

22:         **if** *context* = 5 **then**

23:           **if** user access count $\leq$ resource access count limit **then**

24:             $d_i$= grant

25:             $D$= $d_i$

26:           **else**

27:             $d_i$= deny

28:             $D$= $d_i$

29:           **end if**

30:         **end if**

31:       **end if**

32:    **end if**

33: **end procedure**

---

Algorithm 3 shows the delegation process of our proposed approach. Line 1 of algorithm 3 shows the parameters used in the function delegation() are: delegation id, the one to

whom delegation is transferred, the one who delegates, the delegator role, and the ending time of delegation. The details of the process are shown in Algorithm 3. Line 2 checks if user belongs to role, lines 3 to 4 checks whether the user has previously delegated or not. If the user has previously delegated, the next delegation is not allowed unless the previous delegation is reverted back. if the user has not delegated, line 6 of algorithm 3 shows that the delegation information is inserted to the delegation mapping.

---

**Algorithm 3** Delegation process

---

**Require:** Previous delegation (if any) is reverted

  1: **procedure** *delegation*(del_id, to_user, from_user, del_role, end_time)
  2:     **if** $u_n \in r_n$ **then**
  3:         **if** isdelegated(from_user)=true **then**
  4:             revert
  5:         **else**
  6:             **insert** delegation_information **to** delegation_mapping
  7:         **end if**
  8:     **end if**
  9: **end procedure**

---

## Working of smart contracts

The proposed approach is based on ten smart contracts and their interaction as shown in Fig. 6. The contracts are designed in a modular way to ease the management and minimize the gas cost. These contracts are stored on the Ethereum network blockchain but these contracts can be extended to be stored on other platforms as well. The details of each smart contract are as follows:

- Role contract: This contract manages the users and roles assigned to them. Through this contract, the management can add a role, add users to a role, removed users from a role, delete a role, and update user information.
- Resource contract: This contract manages the resources that are stored on IPFS. The hashes of the resources are be added, removed, and updated through this contract.
- Trust-based contract: This contract checks the reputational score of a user who is member of some role. The reputation is updated based on some factors like the number of views or downloads of the article, the ratings provided by the readers, the citations of the article etc, whose calculation is maintained off chain.
- Cost-based contract: This contract checks the cost constraints that are defined by the policy designer to grant or deny the access to the resource. Cost constraints means a certain cost is defined for a resource that is stored on IPFS.
- Temporal-based contract: This contract checks the temporal constraints that are defined by the policy designer to grant/deny the access to the resource. Temporal constraints means the starting time and end time to access a resource is pre-defined.
- Cardinality-based contract: This contract checks the cardinality constraints that are defined by the policy designer to grant or deny the access to the resource. Cardinality constraints means how many users can get an access of a resource.

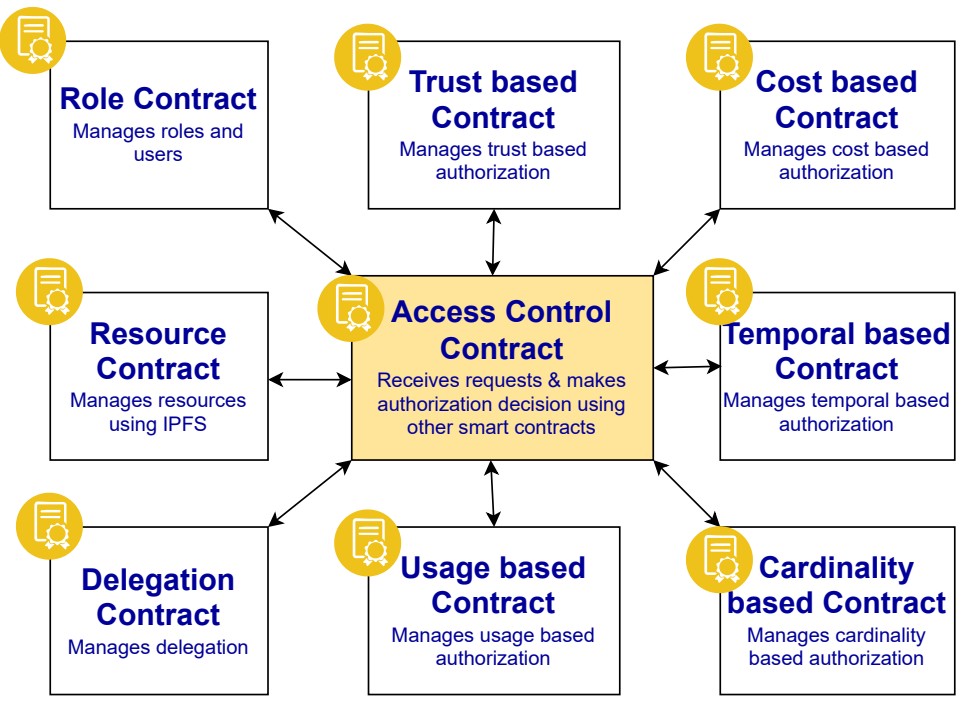

**Figure 6** Smart contracts interaction.

- Usage-based contract: This contract checks the usage constraints that are defined by the policy designer to grant or deny the access to the resource. Usage constraints means how many times a particular user can get an access of a resource at maximum.
- Delegation contract: This contract is invoked when a user performs a delegation request. This contract checks the delegation constraints and then performs the delegation. The delegation information contains the delegation id, the one who is delegating the access privileges, the one to whom the access privileges have been delegated, the starting time, and ending time of the delegation.
- Access control contract: All the contracts' interaction takes place through this contract. This contract receives the access request from the user and sends back the authorization decision of access being granted or denied.

## The delegation process

Our proposed approach as shown in Fig. 5 also handles the context of delegation. Delegation means that the access rights of one user can be delegated to other user. The one to whom the policies are delegated can have the same access rights as that of the delegator *i.e.,* who delegates the access rights. Delegation contract handles the delegation aspects of the our proposed authorization management system. The delegation information in the contract contains the delegation id, the one who is delegating the access privileges, the one to whom the access privileges have been delegated, the starting time, and ending time of the delegation. The contract checks if the delegation exists or not. If the user has already performed delegation, he cannot perform it again unless the previous delegation is reverted

back. Other functionalities of delegation can also be checked through this contract like adding new delegation *i.e.,* pushing the delegation information to the delegation mapping, and retrieval of delegation information by any user.

## IMPLEMENTATION DETAILS

The implementation details of the smart contracts including the data structures used in each of these contracts is provided in this section. The details of smart contracts and their interaction is already presented in section 4.3. We will only present a few core code snippets here, the complete source code of our implementation is available at the following link (https://github.com/Uzmamahar/Thesis-Implementation/blob/main/Complete%20implementation.sol).

The **role contract** manages the users and roles assigned to them. The data structure RoleDetail contains the mapping that maps each user's id to user's address and also contains the variable userscount which is updated whenever the functions adduser() and deleteuser() are called. The other functions used in the contract are updateuser() which is used to update the user information and getuser() which is a view function that is used to view the user's information. This contract can also be invoked when any other contract wants to access the roles or users information.

The **resource contract** is invoked when resource management of any kind is required. The resources are represented through hashes and the details like resource hash, resource name, resource owner, and roles allowed to access a resource are stored in the data structure. The function addResource() adds a resource on the blockchain, addRoleToResource() adds permission to a role that can access a speicfic resource, deleteRoleToResource() removes that permission, deleteResource() removes the resource, updateResourceHash() updates the resource by updating its hash, and getResource() is a view function that displays the resource information. This contract can also be invoked by other contracts when they need to access the resource based authorization decision or fetch any resource information.

Trust based access control model is used to provide authorization decision using **trust-based contract**. The data structure contains two variables i.e, trust limit which is set by the manager, and the reputational/trust score whose mapping is done to each user. Then, RoleTrustManagement is mapped to each role name. Using the function updateTrustLimit(), the manager can update the pre-defined trust limit anytime. The trust score of every user can also be increased or decreased based on the reputational factors. This updating can be done using updateTrust() function. The function getTrust() is a view function which displays the trustscore information of every user when invoked. The core function authorize() defines the trust based authorization policy where user's repute is checked by comparing the reputational/trust score of the user with the trust limit. If the reputational/trust score is greater or equal to the pre-defined trust limit, the access is granted, else denied. This contract can also be invoked by other contracts when they need to access the trust based authorization decision.

The **cost-based contract** is invoked to provide authorization decision based on cost based access control model. This contract checks the cost constraints that are set by the

policy designer for any resource to grant or deny the access to that particular resource. The contract contains mapping that maps the cost of the resource to the resource name. The function updateCost() updates the cost constraints that were previously set by the policy designer. The function getCostforResource() is a view function that displays the cost of resource. The core function authorize() defines the cost based authorization policy that if the cost is greater than or equal to the pre-defined cost of that particular resource, the access is granted, else denied. This contract can be also be invoked by other contracts when they need to access the cost based authorization decision.

The temporal based access control model is used to provide authorization decision by invoking **temporal-based contract**. This contract checks the temporal constraints *i.e.,* the opening/start and closing/end time of a resource, set by the policy designer,to grant or deny the access to that particular resource. The temporal constraints *i.e.,* start time and end time is mapped to every resource name. The function addTemporalConstraints() adds the temporal constraints for a resource, function deleteTemporalConstraints() removes the constraints, and function updateTemporalConstraints() updates the constraints for any resource. The core function authorize() defines the temporal based authoirzation policy that if the user makes an access request at a time that is above the pre-defined starting time and below the pre-defined end time, the access is granted, else denied. This contract can also be invoked by other contracts when they need to access the temporal based authorization decision.

The **cardinality-based contract** is invoked to provide authorization decision based on cardinality based access control model. This contract checks the cardinality constraints *i.e.,* how many users can get an access of a resource. These constraints are defined by the policy designer to grant or deny the access to the resource. The cardinality constraints are mapped to resource names and number of times a resource is accessed is also mapped to resource names. The function addCardinalityConstraint() adds the cardinality constraints, the function removeCardinalityConstraint() removes the constraints and updateCardinalityConstraints()lets the policy designer update them. The core function authorize() defines the cardinality based authoirzation policy that if the number of users granted the access of resource is less than the resource access limit, the access will be granted else denied. This contract can also be invoked by other contracts when they need to access the cardinality based authorization decision.

The usage based access control model is used for the authorization decision by invoking **usage-based contract**. This contract checks the usage constraints *i.e.,* how many times a particular user can get an access of a resource at maximum. The previous user access record of a resource and resource access limit is mapped to every user's address. The function setUserResourceLimit() lets the policy designer set the resource access limit for a user. The core function authorize() defines the usage based authoirzation policy that if number of permissions assigned to a user to access a resource is less than the predefined resource access limit, then the access will be granted else denied.This contract can also be invoked by other contracts when they need to access the usage based authorization decision.

The **delegation contract** is invoked when a user performs a delegation request. For any authorization decision based on five access control models, it is important to check

```
function accessResource(string calldata _role,string calldata _rname)public returns(string memory _reshash, address _rowner){
    require(rolecontract.getUser(_role,msg.sender)!= 0,"user not found in specified role");
    require(!delegationcontract.isDelegated(msg.sender),"user has delegated his permissions");
    if(context==3){
        require(temporalbasedcontract.authorize(_rname),"resource access is not opened yet");
    }else if(context==4){
        require(cardinalitybasedcontract.authorize(_rname),"user reached the access limit");
    }else if(context==5){
        require(usagebasedcontract.authorize(msg.sender),"user reached the access limit");}
    (_reshash,_rowner) = resourcecontract.getResource(_rname); }
```

**Figure 7** **The uploadresource function from the access control contract.**

if the user has delegated his policies to any other user. The data structure for delegation information contains variables *i.e.,* delid, touser, delegatorrole, starttime, and endtime. If a user has already delegated his policies and makes a delegation request the the delegation will be reverted. Delegation information is mapped to every user's address. The function Delegate() checks whether the delegator has previously delegated his policies and this is checked through isDelegated() function. If yes, the function returns that user has already delegated his policies. This delegation request needs to be reverted before making a new delegation request. Another check implemented in the delegate() function is that the end time of delegation should be greater than the current time *i.e.,* the access time, else delegation will not be performed. The RevertDelegation() function returns the rights back to the delegator. The function getdelegation() is a view function that displays the delegation information. This contract can also be invoked when any other contract wants to access the delegation information.

Finally, the **Access Control contract** is the core contract of our proposed methodology. This contract binds all other contracts used in our approach. The access request made by the user is received through this contract and the authorization decision using five access control models is sent back to the user.

Figure 7 shows the function uploadresource() where the Access Control contract verifies all the requirements for a user to upload a resource. In uploadresource() function, the contract first invokes the getuser() function of the rolecontract and checks if the user exists in the specified role. After verification, the contract invokes isDelegated() function of the delegationcontract and checks if the user has delegated his policies to any other user. Then the contract invokes the authorize() function of the trustbasedcontract and checks if the trust score of the user is greater or equal to the pre-defined trust limit. When all the checks are verified, the contract invokes the addResource() function of the resourcecontract and the user can upload the resource by providing his address, the resource hash, resource name, and roles allowed to access the resource.

The function accessPaidResource() is a payable function of the Access Control contract where the authorization decision is made using the cost based access control model. In accessPaidResource() function, the contract first invokes the getuser() function of the rolecontract and checks if the user exists in the specified role. After verification, the contract invokes isDelegated() function of the delegationcontract and checks if the user has delegated his policies to any other user. Then, the contract checks the contexts. If context = 2, it invokes the authorize() function of costbasedccontract and checks the cost constraints

```
function accessResource(string calldata _role,string calldata _rname)public returns(string memory _reshash, address _rowner){
    require(rolecontract.getUser(_role,msg.sender)!= 0,"user not found in specified role");
    require(!delegationcontract.isDelegated(msg.sender),"user has delegated his permissions");
    if(context==3){
        require(temporalbasedcontract.authorize(_rname),"resource access is not opened yet");
    }else if(context==4){
        require(cardinalitybasedcontract.authorize(_rname),"user reached the access limit");
    }else if(context==5){
        require(usagebasedcontract.authorize(msg.sender),"user reached the access limit");}
    (_reshash,_rowner) = resourcecontract.getResource(_rname); }
```

**Figure 8** **The accessresource function from the Access Control contract.**

to access a resource. As it is a payable function, so to return the resource information, we triggered an event paidrssAccess that returns the resource name and other resource details to the user.

The other major function of the Access Control contract is accessResource() as shown in Fig. 8, which take the arguments role and resource name. In accessresource() function, the contract first invokes the getuser() function of the rolecontract and checks if the user exists in the specified role. After verification, the contract invokes isDelegated() function of the delegationcontract and checks if the user has delegated his policies to any other user. Then, the contract checks the contexts. If context = 3, it invokes the authorize() function of temporalbasedccontract and checks the temporal constraints to access a resource. If context = 4, it invokes the authorize() function of cardinalitybasedccontract and checks the cardinality constraints. If context = 5, it invokes the authorize() function of usagebasedccontract and checks the usage constraints. After all the checks are verified, the Access Control contract returns resource hash and other resource details.

## SECURITY ANALYSIS

As blockchain domain has gained popularity, the security concerns of smart contracts have gained wide attention of researchers from all over the world. Smart contracts enable people to make agreements which are conflict free and minimize trust, but they are easily vulnerable to different kinds of attacks. Even a simple smart contract can contain a number of vulnerabilities (*Delmolino et al., 2016*). Since smart contracts deals with cryptocurrencies, the attacks on them result in loss of millions of USD leading to catastrophic financial implications. Examples of attacks include DAO attack in 2016 (*del Castillo, 2016*), parity wallet attack in 2017 (*Palladino, 2017*), and batch overflow attack in 2018 (*Town, 2018*).

As our proposed approach includes smart contracts written in solidity language deployed on Ethereum blockchain platform, we have analysed these smart contracts with respect to common vulnerabilities that exist in the scientific literature. We will first present the list of common vulnerabilities of smart contracts that are applicable to our approach, and then we will present how our proposed approach is secure with respect to these vulnerabilities.

### Common vulnerabilities in smart contracts

We know that our smart contracts' state variables are publicly visible to any valid node. Any security loophole in the smart contracts is publicly noticeable and can be exploited. The list of common vulnerabilities that exist in the literature are:

- Reentrancy: Reentrancy is a vulnerability where the attacker intrudes into the code of the smart contract. When the smart contract invokes the function of an external smart contract, which might be under the control of a malicious user, this external contract might invoke the function of original contract in return causing an invocation loop. The example is that an attacker makes recursive callback of the main function and keeps on executing the withdrawal function of the original contract unless the account balance reaches zero. In 2016, a small bug in the code *i.e.,* developer didn't consider recursive calls, DAO attack took place which resulted in huge economic loss.

- Denial of service attack: When a smart contract calls an external smart contract, the external smart contract can deliberately revert back the transactions of the caller contract and disrupts its execution. In such state the execution of the caller contract is failed and it causes it to be in DoS state.

- Transaction ordering dependence: Every operation performed in a smart contract is a transaction. The order of the execution of these transactions is important for the successful scenario of an operation. For example, if a person has zero tokens in account and he performs two operations *i.e.,* withdraw(500) and deposit(500), the order must be correct for successful execution *i.e.,* first the deposit operation must be executed and then the withdraw operation. If the order is vice versa, the withdraw operation will not successfully execute. A malicious miner can change the order of the transactions that he collected for a block in a way that benefits him.

- Untrusted control flow: When a smart contract calls a function of an external smart contract, the external contract can be an untrusted contract which may be malicious and may result in cases of exploitation.

- Integer overflow and underflow: We use integer types in our smart contracts and solidity has length limit for integers. If a calculation is performed that results in a value above the maximum or below the minimum limit of integers, it will result in a state of overflow or underflow. An attacker may use this vulnerability in a way that benefits him.

- Unprivileged right to storage: This is a vulnerability which is caused by insecure coding in which a function is mandatory to be called by a specific user (for example owner of the contract) but the function to update that specific user (for example owner) has access to every user. Thus, any user can update that specific user in the contract and gets access to his rights.

## Security analysis of proposed smart contracts

We have used the best coding practices considering the common vulnerabilities in order to ensure the security of our proposed smart contracts. In reference to reentrancy vulnerability, our proposed smart contracts do not invoke any external contract for authorization policies management. They invoke the contracts that are internal to our system and even do not change the state of the contract after invoking contracts' functions. For reference, we have shown the accessresource() function from Access Control contract in Fig. 9.

Regarding denial of service vulnerability, our proposed smart contracts is neither invoked nor invokes any external smart contract. Our contracts utilize pre-defined functionality of the contracts internal to our system, which are not malicious. In reference to transaction

```
function accessResource(string calldata _role,string calldata _rname)public
returns(string memory _reshash, address _rowner){
    require(rolecontract.getUser(_role,msg.sender)!= 0,"user not found in specified role");
    require(!delegationcontract.isDelegated(msg.sender),"user has delegated his permissions");
    if(context==3){
        require(temporalbasedcontract.authorize(_rname),"resource access is not opened yet");
    }else if(context==4){
        require(cardinalitybasedcontract.authorize(_rname),"user reached the access limit");
    }else if(context==5){
        require(usagebasedcontract.authorize(msg.sender),"user reached the access limit");}
    (_reshash,_rowner) = resourcecontract.getResource(_rname); }
```

**Figure 9** **The accessresource function from the Access Control contract.**

ordering dependency, our proposed smart contracts do not provide multiple authorization decisions with respect to one scenario at a same time. For example, the authorization decision cannot be deny and allow at the same time for a particular access resource scenario. It can be either be deny or allow regardless of the transaction order.

In regards to untrusted control flow, our proposed approach does not invoke any external contract. Therefore, this vulnerability is not applicable to our approach. Our contracts do not alter its flow after calling other contracts of the system. As far as arithmetic vulnerability *i.e.,* integer overflow and underflow is concerned, it has already been handled since solidity 0.8.0 is released. The EVM, by default, reverts the transaction if integer overflows or underflows. To deal with unprivileged right to storage, most of the functions of our smart contracts that change the state are only accessible to the owner. The access to update the owner's address is allowed to the owner only. This can be seen in the modifier function from the Role contract in Fig. 10.

## PERFORMANCE EVALUATION

In this section, we present the performance evaluation results of our proposed approach. We have written and deployed our smart contracts using Ethereum framework, and also used Ganache v2.5.4.0 for the deployment purpose. We have also used Metamask wallet to deploy our smart contracts on Rinkeby test network which is an Ethereum test network used for development testing before deploying on Ethereum main network. For the interaction with smart contracts, we have used Truffle v5.5.7 and Remix IDE. The programming language used to write the smart contracts is Solidity and the compiler version used is 0.8.7+commit.e28d00a7.

Every transaction that is Ethereum based generates a fee known as gas cost. The existing work in scientific literature is mostly based on time as a parameter to evaluate their approaches which in our view is not a good measure to evaluate the performance. The reason is that, in practice, the transaction propagation time is complex as the transaction confirmation time depends on transaction congestion in blocks. Users can manipulate time by paying a competitive gas price and the transaction confirmation time will be decreased for them. Hence, time cannot be taken as a good parameter for measuring performance.

For the performance evaluation of our approach which is based on smart contracts, we have taken gas cost as a parameter as it depends on the data structure used in writing

```
contract RoleContract{
    address private owner;
    constructor(){
        owner = msg.sender;  }
    modifier onlyowner() {
        require(msg.sender == owner, "Not owner");
        _;}
    function changeowner(address _owner) public onlyowner{
        owner=_owner;
    }
    struct RoleDetail{ ···
    }
    mapping(string => RoleDetail) roles;
    function addUser(string calldata _role,address _user,uint256 _uid) public onlyowner {
        roles[_role].users[_user] =_uid;
        roles[_role].userscount++; }
```

**Figure 10** **The modifier function from the role contract.**

these smart contracts and can be an effective measure to evaluate the performance. The details of the performance evaluation and results is available at the following link (https://github.com/Uzmamahar/Performance-evaluation-/blob/main/Performance%20evaluation%20results.xlsx). To calculate the gas cost, we have divided it into two categories *i.e.,* deployment gas cost of the smart contracts and evaluation gas cost of the functions of smart contracts which are used for performing the authorization management process. The gas cost for deploying the smart contracts can be expressed as:

$$Gas_{dep} = Execution\ Cost + Fixed\ Cost_{dep} + Code\ Cost.$$

Fixed cost above represents the fixed cost of deployment of a smart contract which is 32,000 gas units as per the Ethereum's yellow paper. The code cost is the gas cost of the actual code or lines of code needed to write the smart contract and execution cost is the initialization cost of the smart contract and the cost to run the constructor instructions. The execution cost of almost all the smart contracts is same for our proposed approach *i.e.,* 20117 gas units, except the delegation contract and access control contract which is 51 gas units and 181,134 gas units respectively. Our usage-based contract has the minimum code cost *i.e.,* 283,850 gas units and Access Control contract has the maximum code cost which is 1,438,854 because it contains the core logic of the authorization management and hence has more byte-code stored on blockchain. Similarly, the total deployment cost is also minimum for usage-based contract and maximum for Access control contract. The deployment cost of our smart contracts at Ganache ranges from 334,727 gas units of usage-based contract to 1,640,276 gas units of Access Control contract. The details of deployment cost of every contract are shown in Fig. 11. We have taken both the gas cost values of remix and Ganache and have averaged them out for better accuracy of results. The average deployment gas cost of our proposed framework is shown in Fig. 12.

We have also calculated the evaluation gas cost of our smart contracts which is used when the functions are invoked for authorization decisions. The evaluation cost consists of a fixed cost for every smart contract function which is 21,000 gas units as per the Ethereum

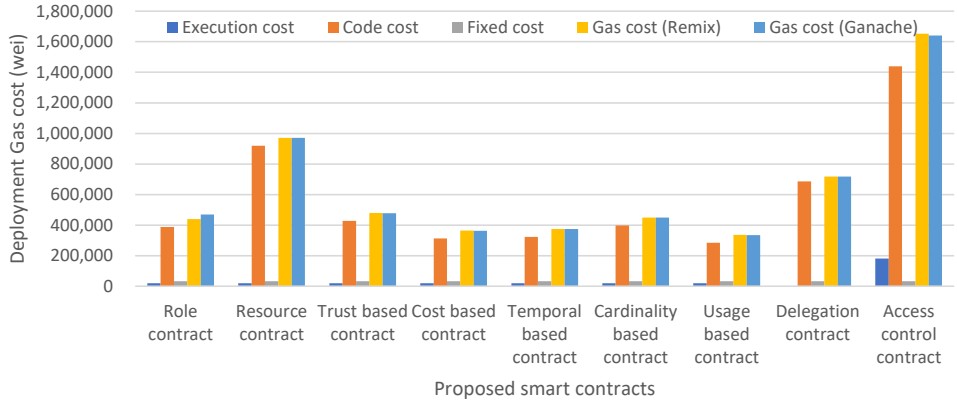

**Figure 11 Deployment gas cost of our proposed framework.**

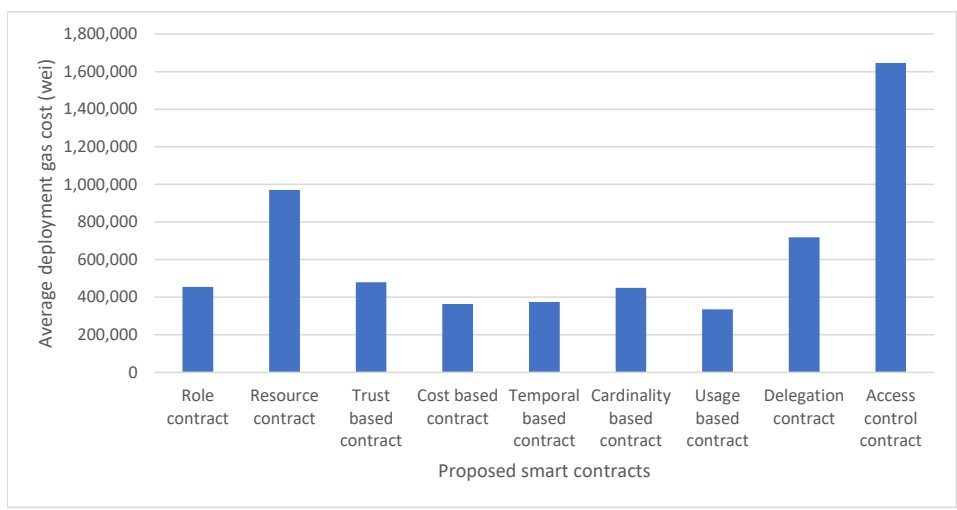

**Figure 12 Average deployment gas cost using remix and ganache.**

yellow paper and execution cost which varies from function to function. The details of the evaluation gas cost for our proposed framework are shown in Fig. 13. The average evaluation gas cost of our proposed framework, taken from taking average of remix and ganache gas cost, is shown in Fig. 14. The gas cost for evaluating the smart contracts can be expressed as:

$$Gas_{eval} = Execution\ Cost + Fixed\ Cost_{eval}$$

Figure 14 shows that the function uploadresource() of Access Control smart contract has the highest evaluation gas cost as to deliver an authorization decision of 'access-grant' to the user, it first invokes the role contract to see if the role exists and the user is found in the specified role, then it invokes the delegation contract to check if the user has delegated his policies to any other user or not, and finally invokes the trust based contract to check the trust score of the user to upload resource. That is why it consumes the maximum gas

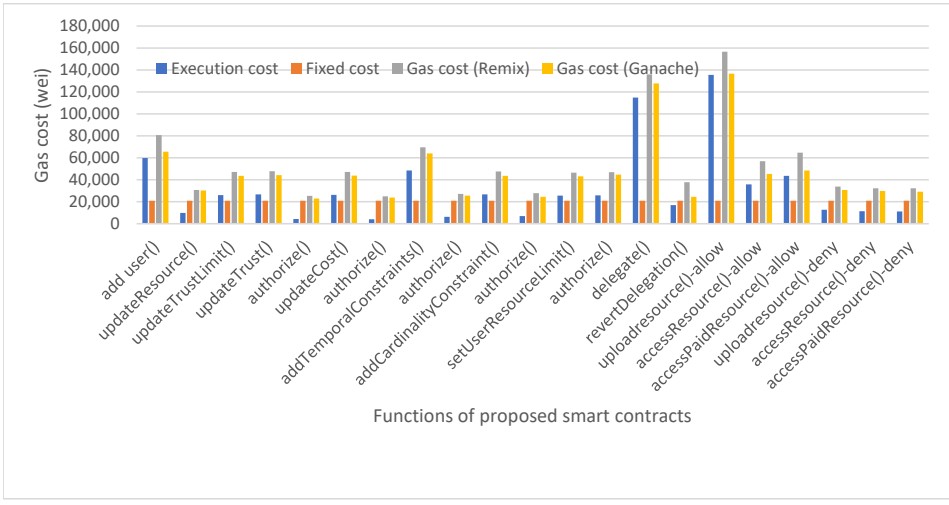

**Figure 13  Evaluation gas cost of our proposed framework.**

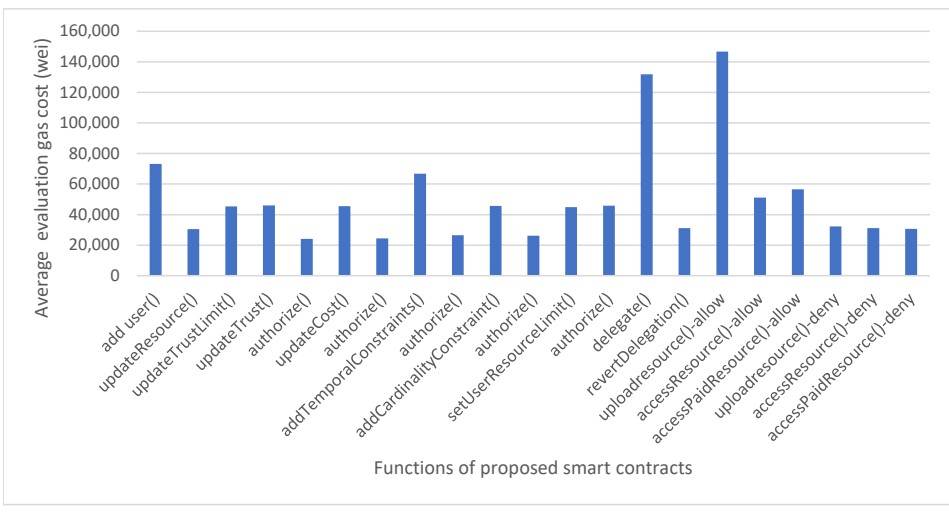

**Figure 14  Average evaluation gas cost using remix and ganache.**

cost *i.e.,* 146,603 gas units. But we can see that the uploadresource() function in case of authorization decision as access-deny consumes less gas cost because first it checks the role contract and if the user does not exist in the specified role, it does not invoke the rest contracts and results as access denied.

## RESULTS AND DISCUSSION

A part of our approach can be compared to *Putra et al. (2021b)* in which the authors have worked on access control mechanism using trust-based access model. A full comparison cannot be done as the article does not address the advance aspects of authorization

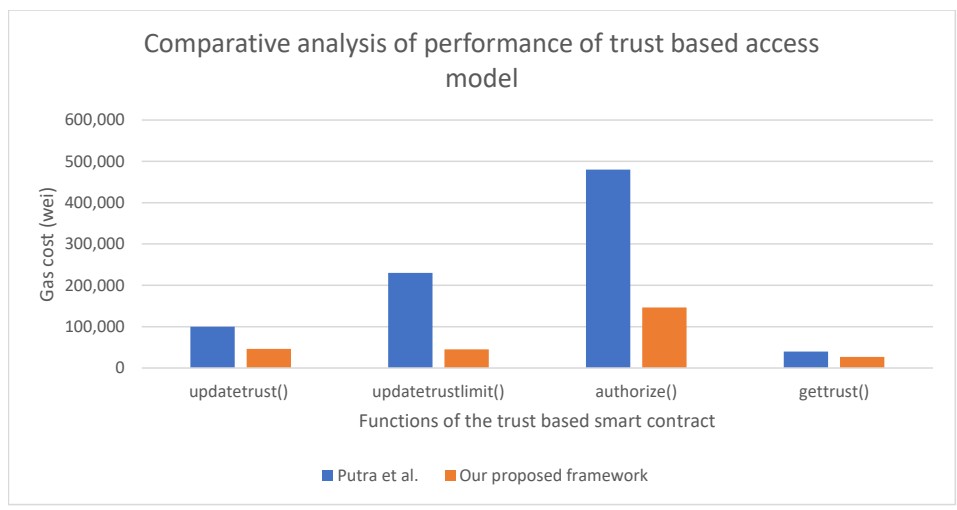

**Figure 15** **Performance of our proposed approach in comparison to *Putra et al. (2021b)*.**

management as addressed in our article. Their approach does not focus on the other four access control models and does not address delegation. We have compared the functions of trust-based access control model with *Putra et al. (2021b)* as shown in Fig. 15. It can be seen that our proposed trust-based contract consumes 60.4% less gas cost as compared to *Putra et al. (2021b)*.

Our work can also be partially compared to *Sherazi et al. (2022)* with respect to role contract, resource contract, management contract which is partially similar to our proposed access control contract, and delegation contract. The five access control models used in our approach are not covered in *Sherazi et al. (2022)*, hence, a full comparison is not possible. The comparison is shown in Fig. 16, where we can see that our proposed smart contracts consume 59.2% less gas cost in comparison to the approach presented in *Sherazi et al. (2022)*. This proves that our proposed approach perform authorization management process more efficiently with minimal gas consumption.

## CONCLUSION AND FUTURE WORK

In this article, we have presented an authorization management framework TTECCDU that comprises of multi access control models *i.e.,* trust-based, cost-based, temporal-based, cardinality-based and usage-based access control model. TTECCDU framework provides a strong and expressive authorization mechanism for large scale distributed systems. Our approach is based on role based access control model which comprises of several smart contracts to provide the authorization decision as access allowed or denied. We have also handled the delegation context to to prevent faulty authorization decisions. TTECCDU is implemented using smart contracts which are written in a modular form so that they are easily manageable. For implementation of our approach, Ethereum framework and the programming language *i.e.,* Solidity was used. Remix and Ganache was used for the deployment and evaluation of these contracts. Performance evaluation results show that

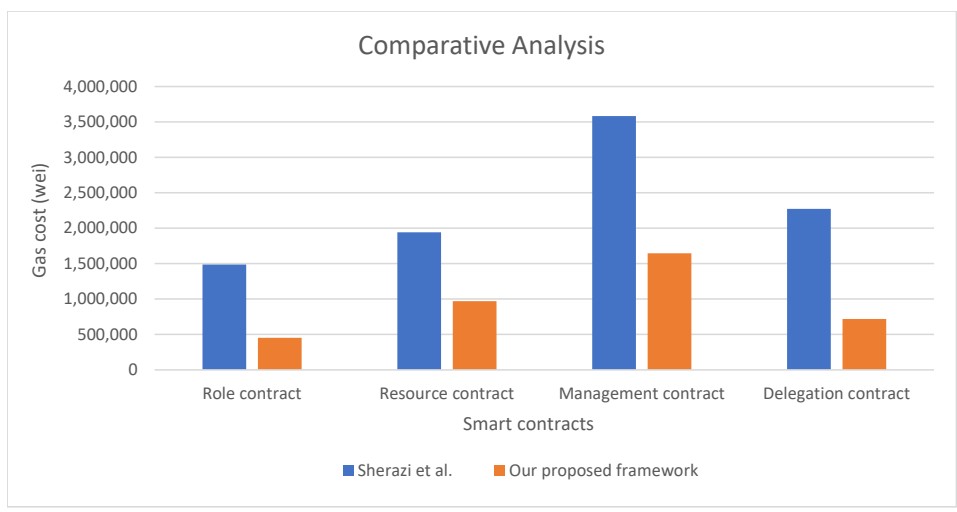

**Figure 16** **Performance of our proposed approach in comparison to** *Sherazi et al. (2022)*.

our smart contracts are written in an optimized manner which consume 60.4% less gas cost when the trust-based access is compared and 59.2% less gas cost when other proposed smart contracts are compared to the existing approaches. Our approach is also more secure as evident from the security analysis of our framework.

In the future, we intend to explore some more access control models that enhance the security and further reduce the gas cost. We also intend to implement some advance delegation aspects.

### Funding
The authors received no funding for this work.

### Competing Interests
Muhammad Aleem is an Academic Editor for PeerJ Computer Science. Uzma Mahar is employed by National University of Computer and Emerging Sciences Islamabad Pakistan. Ehtesham Zahoor is employed by Educative, Inc. The authors have no competing interests.

### Author Contributions
- Uzma Mahar conceived and designed the experiments, performed the experiments, analyzed the data, performed the computation work, prepared figures and/or tables, and approved the final draft.
- Muhammad Aleem conceived and designed the experiments, performed the experiments, analyzed the data, prepared figures and/or tables, authored or reviewed drafts of the article, and approved the final draft.

- Ehtesham Zahoor conceived and designed the experiments, performed the computation work, prepared figures and/or tables, authored or reviewed drafts of the article, and approved the final draft.

## Data Availability

The Implementation of Authorization policies are available at Github and Zenodo: https://github.com/Uzmamahar/Thesis-Implementation.

Uzmamahar. (2022). Uzmamahar/Thesis-Implementation: release (0.1.0). Zenodo. https://doi.org/10.5281/zenodo.7386616.

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
