# Peer review of "TTECCDU: a blockchain-based approach for expressive authorization management"

_PeerJ Computer Science, doi:10.7717/peerj-cs.1212_

## Round 0.1 · original submission · Major Revisions

Please carefully address the reviewer's comments. Both reviewers have provided valuable comments. The paper requires several amendments and corrections related to experimental design and the validity of the findings.

Reviewer 1 ·

Basic reporting

The manuscript presented a block-chain based technique for expressive authorization management. This work is interesting that proposed the important authorization factors like trust, cost, and cardinality part of a unified framework. However, the following comments should be addressed if the work is being considered for publication:
• In Abstract: a brief of the existing available work should be provided (maybe in 2 to 3 lines of the text).
• Line 62, the approach name “TTECCDU” has not been reflected in the abstract and title. The authors may add to these too.
• The core contribution points should be concise. Especially contribution 4 looks like a full paragraph which is incorrect approach.
• The first contribution point needs a lot of improvements. Combining multiple trust factors should not be the core objective. What the author tried to achieve by combining those should have been highlighted.
• Introduction should extend the text with the rationale of combining multi-factors for authorization purposes.
• The related work section is written well and the research gap discussion is comprehensive.
• Table 1 caption should be at the top.
• In Table 1, listed several authorization types as letters. The authors should add a separate table to provide a single-line explanation of the authorization types mentioned as letters.
• Discussion of Figure 3 should include the line numbers while referring to the code lines.
• The NEWS case study should be concise. The discussion at the moment is too extensive.
• In Figure 4, the sequencing of the mentioned steps is not very clear the figure may be re-structured and re-drawn to convey the idea in a clear and concise manner.
• Algorithm 1 and Algorithm 2, and Algorithm 3 should be revised to include the system model terms instead of defining the new ones of the text style variables. The algorithm should be relatable to the given system model and use terminologies.
• Section 4.3, mainly presents the background information; therefore it should be concise and reduced.
• Figure 7 and Figure 8, and Figure 11 shows the code that has different font size. This should be adjusted.
• The Section implementation details are too large, the implementation files could be uploaded to the public repository and the images related to the code should be removed. The discussion can be further concise.
• The experimental evaluation section contains figures that have poor quality. The authors should replace with vector graphics images (All figures in the paper should be replaced accordingly, especially the experimental evaluation Section).
• The results discussions should be added to the paper (after the experimental evaluation section). The authors should present the technical discussion about the overall achievements of the work considering the evaluation of the results.

Experimental design

The manuscript presented a block-chain based technique for expressive authorization management. This work is interesting that proposed the important authorization factors like trust, cost, and cardinality part of a unified framework. However, the following comments should be addressed if the work is being considered for publication:
• In Abstract: a brief of the existing available work should be provided (maybe in 2 to 3 lines of the text).
• Line 62, the approach name “TTECCDU” has not been reflected in the abstract and title. The authors may add to these too.
• The core contribution points should be concise. Especially contribution 4 looks like a full paragraph which is incorrect approach.
• The first contribution point needs a lot of improvements. Combining multiple trust factors should not be the core objective. What the author tried to achieve by combining those should have been highlighted.
• Introduction should extend the text with the rationale of combining multi-factors for authorization purposes.
• The related work section is written well and the research gap discussion is comprehensive.
• Table 1 caption should be at the top.
• In Table 1, listed several authorization types as letters. The authors should add a separate table to provide a single-line explanation of the authorization types mentioned as letters.
• Discussion of Figure 3 should include the line numbers while referring to the code lines.
• The NEWS case study should be concise. The discussion at the moment is too extensive.
• In Figure 4, the sequencing of the mentioned steps is not very clear the figure may be re-structured and re-drawn to convey the idea in a clear and concise manner.
• Algorithm 1 and Algorithm 2, and Algorithm 3 should be revised to include the system model terms instead of defining the new ones of the text style variables. The algorithm should be relatable to the given system model and use terminologies.
• Section 4.3, mainly presents the background information; therefore it should be concise and reduced.
• Figure 7 and Figure 8, and Figure 11 shows the code that has different font size. This should be adjusted.
• The Section implementation details are too large, the implementation files could be uploaded to the public repository and the images related to the code should be removed. The discussion can be further concise.
• The experimental evaluation section contains figures that have poor quality. The authors should replace with vector graphics images (All figures in the paper should be replaced accordingly, especially the experimental evaluation Section).
• The results discussions should be added to the paper (after the experimental evaluation section). The authors should present the technical discussion about the overall achievements of the work considering the evaluation of the results.

Validity of the findings

The manuscript presented a block-chain based technique for expressive authorization management. This work is interesting that proposed the important authorization factors like trust, cost, and cardinality part of a unified framework. However, the following comments should be addressed if the work is being considered for publication:
• In Abstract: a brief of the existing available work should be provided (maybe in 2 to 3 lines of the text).
• Line 62, the approach name “TTECCDU” has not been reflected in the abstract and title. The authors may add to these too.
• The core contribution points should be concise. Especially contribution 4 looks like a full paragraph which is incorrect approach.
• The first contribution point needs a lot of improvements. Combining multiple trust factors should not be the core objective. What the author tried to achieve by combining those should have been highlighted.
• Introduction should extend the text with the rationale of combining multi-factors for authorization purposes.
• The related work section is written well and the research gap discussion is comprehensive.
• Table 1 caption should be at the top.
• In Table 1, listed several authorization types as letters. The authors should add a separate table to provide a single-line explanation of the authorization types mentioned as letters.
• Discussion of Figure 3 should include the line numbers while referring to the code lines.
• The NEWS case study should be concise. The discussion at the moment is too extensive.
• In Figure 4, the sequencing of the mentioned steps is not very clear the figure may be re-structured and re-drawn to convey the idea in a clear and concise manner.
• Algorithm 1 and Algorithm 2, and Algorithm 3 should be revised to include the system model terms instead of defining the new ones of the text style variables. The algorithm should be relatable to the given system model and use terminologies.
• Section 4.3, mainly presents the background information; therefore it should be concise and reduced.
• Figure 7 and Figure 8, and Figure 11 shows the code that has different font size. This should be adjusted.
• The Section implementation details are too large, the implementation files could be uploaded to the public repository and the images related to the code should be removed. The discussion can be further concise.
• The experimental evaluation section contains figures that have poor quality. The authors should replace with vector graphics images (All figures in the paper should be replaced accordingly, especially the experimental evaluation Section).
• The results discussions should be added to the paper (after the experimental evaluation section). The authors should present the technical discussion about the overall achievements of the work considering the evaluation of the results.

Additional comments

The manuscript presented a block-chain based technique for expressive authorization management. This work is interesting that proposed the important authorization factors like trust, cost, and cardinality part of a unified framework. However, the following comments should be addressed if the work is being considered for publication:
• In Abstract: a brief of the existing available work should be provided (maybe in 2 to 3 lines of the text).
• Line 62, the approach name “TTECCDU” has not been reflected in the abstract and title. The authors may add to these too.
• The core contribution points should be concise. Especially contribution 4 looks like a full paragraph which is incorrect approach.
• The first contribution point needs a lot of improvements. Combining multiple trust factors should not be the core objective. What the author tried to achieve by combining those should have been highlighted.
• Introduction should extend the text with the rationale of combining multi-factors for authorization purposes.
• The related work section is written well and the research gap discussion is comprehensive.
• Table 1 caption should be at the top.
• In Table 1, listed several authorization types as letters. The authors should add a separate table to provide a single-line explanation of the authorization types mentioned as letters.
• Discussion of Figure 3 should include the line numbers while referring to the code lines.
• The NEWS case study should be concise. The discussion at the moment is too extensive.
• In Figure 4, the sequencing of the mentioned steps is not very clear the figure may be re-structured and re-drawn to convey the idea in a clear and concise manner.
• Algorithm 1 and Algorithm 2, and Algorithm 3 should be revised to include the system model terms instead of defining the new ones of the text style variables. The algorithm should be relatable to the given system model and use terminologies.
• Section 4.3, mainly presents the background information; therefore it should be concise and reduced.
• Figure 7 and Figure 8, and Figure 11 shows the code that has different font size. This should be adjusted.
• The Section implementation details are too large, the implementation files could be uploaded to the public repository and the images related to the code should be removed. The discussion can be further concise.
• The experimental evaluation section contains figures that have poor quality. The authors should replace with vector graphics images (All figures in the paper should be replaced accordingly, especially the experimental evaluation Section).
• The results discussions should be added to the paper (after the experimental evaluation section). The authors should present the technical discussion about the overall achievements of the work considering the evaluation of the results.

Reviewer 2 ·

Basic reporting

The paper is well written. However, there are several technical typos that need to be fixed:
1> For instance, in section 4.1, U_1 \in r_1 should be u_1 \in r_1 and so on

The references to the literature are also sufficient.

All the tables and figures are well presented.

All the details about the approach and experiment are mentioned.

The results are clearly discussed.

Experimental design

The research is within the scope of the journal.

The research questions are well defined and discussed.

The experiment is well organized.

The approach is explained in adequate detail.

Validity of the findings

The approach is novel, however, the algorithms should be combined to show the overall approach because most of the algorithms have duplicate computations and thus do not make sense.

It is not clear if the data used in the experiment is public. I suggest making the data public and providing the link in the paper or explaining the data features and basis otherwise.

The conclusions are well stated.

---

## Round 0.2 · accepted · Accept

The authors have addressed all of the reviewers' comments.

Reviewer 1 ·

Basic reporting

The comments has been addressed and i have no further comments.

Experimental design

The comments has been addressed and i have no further comments.

Validity of the findings

The comments has been addressed and i have no further comments.

Additional comments

The comments has been addressed and i have no further comments.

Reviewer 2 ·

Basic reporting

All of my previous comments have been addressed.

Experimental design

All of my previous comments have been addressed.

Validity of the findings

All of my previous comments have been addressed.

Additional comments

All of my previous comments have been addressed.